# Outcomes of decitabine treatment for newly diagnosed acute myeloid leukemia in older adults

**Kwai Han Yoo[1], Jinhyun Cho[2], Boram Han[3], Se Hyung Kim[4], Dong-Yeop Shin[5], Junshik Hong[5]\*, Hawk Kim[1], Hyo Jung Kim[3], Dae young Zang[3], Sung-Soo Yoon[5], Jong-Youl Jin[6], Jae Hoon Lee[1], Dae-Sik Hong[4], Seong Kyu Park[4]\***

**1** Division of Hematology, Department of Internal Medicine, Gachon University Gil Medical Center, Gachon University College of Medicine, Incheon, Korea, **2** Division of Hematology-Oncology, Department of Internal Medicine, Inha University Hospital, Inha University School of Medicine, Incheon, Korea, **3** Division of Hematology/Oncology, Department of Internal Medicine, Hallym University Sacred Heart Hospital, Hallym University College of Medicine, Anyang, Korea, **4** Division of Hemato-Oncology, Department of Internal Medicine, Soonchunhyang University Bucheon Hospital, Bucheon, Korea, **5** Department of Internal Medicine, Seoul National University College of Medicine, Seoul National University Hospital, Seoul, Korea, **6** Division of Hemato-Oncology, Department of Internal Medicine, Bucheon St. Mary's Hospital, Bucheon, Korea

\* alertjun@hanmail.net (JH); skpark@schmc.ac.kr (SKP)

# Abstract

**Data Availability Statement:** There are no ethical or legal restrictions on sharing a de-identified data set. We shared the data set via figshare: (https://

## Purpose

We evaluated the outcomes of decitabine as first-line treatment in older patients with acute myeloid leukemia (AML) and investigated the predictors, including a baseline mini nutritional assessment short form (MNA-SF) score, of response and survival.

## Patients and methods

Between 2010 and 2018, 96 AML patients aged 65 and above who received decitabine treatment at 6 centers in Korea were retrospectively evaluated. Response rates, hematologic improvements (HI), progression-free survival (PFS), and overall survival (OS) were analyzed.

## Results

The median age at diagnosis was 73.9 years, and the median number of decitabine treatments administered to the patients was 4 (range, 1−29). Of 85 patients, 15 patients (17.6%) achieved complete remission (CR) or CR with incomplete blood count recovery. Twelve patients (14.1%) showed partial remission (PR), and 18 (21.2%) demonstrated HI without an objective response. The median PFS and OS were 7.0 (95% confidence interval [CI], 4.9 −9.0) and 10.6 (95% CI, 7.7−13.5%) months, respectively. In multivariate analyses, MNA-SF score ≥ 8 and the absence of peripheral blood (PB) blasts were significant predictors for improved PFS and OS.

figshare.com/articles/dataset/Sharing_data/
12038748) (10.6084/m9.figshare.12038748).

**Funding:** KHY was supported by the Gachon
University Research Grants in 2018 (GCU-2018-
5260). Gachon University (https://www.gachon.ac.
kr) The funder had no role in study design, data
collection and analysis, decision to publish, or
preparation of the manuscript.

**Competing interests:** The authors have declared
that no competing interests exist.

## Conclusions

For older patients with newly diagnosed AML, a high MNA-SF score and the absence of PB
blasts were independently associated with improved survival.

## Introduction

Acute myeloid leukemia (AML) is the most common acute leukemia in adults and is charac-
terized by clonal expansion of myeloid blasts resulting from somatic mutations in a primitive
multipotential hematopoietic cell [1]. The median age of AML patients at diagnosis was
reported to be around 70 years [2], and treatment strategies and outcomes were significantly
influenced by patients' age [3]. Treatment of AML in older adults encounters two major obsta-
cles, therapeutic resistance of the disease and patients' intolerance to intensive chemotherapy
[4]. Thus, the rate of remission induction chemotherapy in older AML patients was reduced,
and other therapeutic options such as hypomethylating agents, low dose cytarabine, or best
supportive care with oral cytostatic drugs could be introduced to the treatment plan [5]. There-
fore, the treatment of AML in older patients requires a geriatric approach.

Decitabine, a hypomethylating agent inhibiting DNA methyltransferase, first demonstrated
its therapeutic efficacy for myelodysplastic syndrome (MDS) [6, 7], and it was also investigated
for its uses in the treatment of AML. Decitabine was well tolerated and demonstrated a 26%
response rate in a multicenter phase II trial for older AML patients who were unfit for induc-
tion chemotherapy [8]. In a phase III study conducted in 2012, in which the efficacy of decita-
bine was compared to that of low dose cytarabine or supportive care for older patients with
newly diagnosed AML, decitabine improved response rates and showed a benefit in overall
survival (OS) in a post-hoc analysis [9]. Based on these studies, a marketing authorization
valid throughout the European Union (EU) was issued for decitabine for the treatment of
adult patients aged 65 years and older with newly diagnosed de novo or secondary AML who
are not candidates for standard induction chemotherapy in 2012. Decitabine was also
approved by the Korean Food and Drug Administration (KFDA) in 2013, while the United
States Food and Drug Administration (USFDA) did not approve decitabine for the treatment
of newly diagnosed AML. There is still controversy, but it has been widely used for initial treat-
ment of AML in patients aged 65 years or above.

Recently, several retrospective studies of decitabine in older patients with AML have been
reported [10–12]. However, predictors for response to decitabine, duration of response, and
survival have not been well elucidated. Thus, we conducted a multicenter retrospective study
of decitabine treatment in older patients with newly diagnosed AML.

## Patients and methods

### Patients

This study was approved by the institutional review board of Seoul National University Hospi-
tal (IRB No: H-1802-018-919). Informed consent was waived because of the retrospective
nature of the study and the analysis used anonymous clinical data. All data were fully anon-
ymized prior to access for analysis. The access to patients' medical records was made between
January 2017 and June 2019.

Older patients with newly diagnosed AML from 6 institutions in Korea were included in
this study. The inclusion criteria were as follows: (a) patients diagnosed with AML according

to the 2008 World Health Organization (WHO) criteria; (b) age 65 years or above at the time of initial diagnosis; (c) patients who were not eligible for standard induction remission chemotherapy and who received decitabine as first-line treatment between 2010 and 2018; and (d) patients with complete data regarding baseline characteristics and treatment outcomes. Patients with acute promyelocytic leukemia (APL), with central nervous system involvement of AML, and relapsed AML after prior systemic chemotherapy were excluded from the study.

Data of patient demographics, baseline characteristics, and screening parameters for a mini nutritional assessment short form (MNA-SF), including decline in food intake, weight loss, mobility, neuropsychological problems, and body mass index [13], were obtained by reviewing electronic medical records (EMRs). All institutions participating in this study had acquired all MNA-SF-related indicators at the time of diagnosis of AML, through hospitalization records, baseline nursing records, and nutritional records assessed by on-site nutritionists. AML was categorized as AML with recurrent genetic abnormalities, AML with myelodysplasia-related changes, therapy-related myeloid neoplasms, and AML, not otherwise specified according to the 2016 revision of the World Health Organization (WHO) classification of myeloid neoplasms and acute leukemia [14]. Cytogenetic risks were re-classified by the 2017 European LeukemiaNet (ELN) risk stratification [15]. This study was approved by the institutional review board.

## Treatment and evaluation

Patients received decitabine 20 mg/m$^2$ per day for 5 consecutive days every 4 weeks. Treatment was continued until death, treatment failure, unacceptable toxicities, or lack of clinical benefit.

Bone marrow (BM) biopsies and aspirates were not mandatory if treatment failure was strongly suggested (i.e., new presence of leukemic blasts in peripheral blood [PB] or lack or loss of hematologic improvement [HI] during treatment) considering the mechanism of action of decitabine and delayed responses different from intensive chemotherapy, but they were otherwise performed within 4–6 cycles of decitabine treatment for response evaluation. Treatment response was evaluated according to the 2003 revised International Working Group (IWG) AML criteria [16]. HIs in the 3 hematopoietic lineages were assessed in PB according to the 2006 IWG response criteria for myelodysplastic syndrome [17]. Adverse events (AEs) ≥ grade 3 were collected, especially focusing on infectious complications. Infectious complications included any bacterial, viral, fungal, and miscellaneous infection such as *Pneumocystis jiroveci* during decitabine treatment.

## Statistical analysis

Differences between groups were assessed using the Student's t-test for continuous variables. Comparison of dichotomous or categorical variables was based on the Pearson's chi-squared test or Fisher's exact test. Progression-free survival (PFS) and OS were measured from the initiation of decitabine treatment to progressive disease (PD) and death by any cause, respectively. The Kaplan-Meier method was used to evaluate PFS and OS. PFS and OS were compared using a log-rank test in univariate analysis. Variables which were statistically significant in univariate analysis of PFS and OS ($P < .05$) were used as covariates in multivariate analysis. Multivariate Cox proportional hazards model assessed the association of covariates and PFS and OS. All *P*-values were 2-tailed. *P*-values less than 0.05 were considered significant. All data were analyzed using the Statistical Package for the Social Sciences software (IBM® SPSS® statistics, version 23.0).

## Results

### Patient characteristics

A total of 96 AML patients satisfied the inclusion criteria and their data were analyzed. The majority of patients were male (n = 57, 59.4%), and the median age at diagnosis was 73.9 years (range 65−91 years). Forty-six patients (47.9%) had Eastern Cooperative Oncology Group (ECOG) performance status of 0 or 1, and 50 patients (52.1%) had ECOG performance status ≥ 2. The median body mass index (BMI) and MNA-SF score were 23.2 (range, 16.3 −32.2) kg/m$^2$ and 9 (range 4−13), respectively. Detailed baseline characteristics are given in Table 1.

### Treatment responses and adverse events

Treatment outcomes are given in Table 2. A total of 550 cycles of decitabine were administered, and the median number of decitabine treatments received by patients was 4 (range, 1−29). Of 85 patients who were evaluable for treatment response, 11 (12.9%) achieved complete remission (CR) and 4 patients (4.7%) had CR with incomplete blood count recovery (CRi). Twelve patients (14.1%) showed partial remission (PR), and 18 patients (21.2%) who did not achieve an objective response demonstrated hematologic improvement (HI) in PB. Thus, the clinical benefit rate (CR + CRi + PR + HI only) was 52.9% (45/ 85). Regardless of achieving an objective response, 45 patients (45/90, 50.0%) showed HI in absolute neutrophil count (29/78, 37.2%), hemoglobin (32/78, 41.0%) and/or platelet count (28/77, 36.4%). Forty-two patients (43.8%) experienced AEs ≥ grade 3, and most of them were infectious complications (n = 36, 37.5%). Bacterial infection was most common (n = 31, 32.3%), followed by fungal infection (n = 6, 6.3%). Twelve patients (12.5%) died during the induction period due to infection (n = 9, 9.4%), rapidly progressive disease (n = 2, 2.1%), and thrombosis (n = 1, 1.0%). Twenty-five patients (26.0%) discontinued decitabine without treatment failure, mainly due to deteriorated performance (n = 21, 21.8%).

Upon comparison of the decitabine treatment responders (CR, CRi, or PR, n = 27) and non-responders (n = 69), it was found that the responder group included more male patients (77.8% vs. 52.2%, *P* = .022) and that patients in the responder group had fewer blasts in their BM (median 43% vs. 62%, *P* = .015). On the contrary, PB blasts were more frequently exist in the non-responders than in the responders (62% vs. 44.4%, *P* = .022, Table 3).

### Survival outcomes and predictors of measures of survival

The median PFS and OS were 7.0 (95% confidence interval [CI], 4.9−9.0) and 10.6 (95% CI, 7.7−13.5) months, retrospectively (Fig 1). As determined by univariate subgroup analyses for PFS, age ≤ 75 years, ECOG performance status 0 or 1, favorable or intermediate cytogenetic risk group, the absence of PB blasts, and an MNA-SF score ≥ 8 (at risk to normal) were all associated with improved survival (S1 Fig). Age, performance status, the absence of PB blasts, and the MNA-SF score were also associated prolonged OS in univariate analysis. Patients in the favorable or intermediate cytogenetic risk group showed longer OS than patients in the poor risk group, though this result was not statistically significant (S2 Fig). The percentage of BM blasts (cutoff value 30% and/or 50%) was not associated with either PFS or OS. In the multivariate analysis, 5 covariates (age, ECOG performance status, MNA-SF score, the absence of PB blasts, and cytogenetic risk) were used equally for Cox regression of PFS and OS. An MNA-SF score ≥ 8 and the absence of PB blasts were the most significant predictors for both

**Table 1. Patient characteristics.**

| | |
|---|---|
| **Age, years** | |
| Median (range) | 73.9 (65−91) |
| **Sex** | |
| Male | 57 (59.4%) |
| **Eastern Cooperative Oncology Group (ECOG) performance status** | |
| 0−1 | 46 (47.9%) |
| 2−4 | 50 (52.1%) |
| **Body mass index (BMI)** | |
| Median (range) | 23.2 (16.3−32.2) |
| **Mini nutritional assessment short from (MNA-SF) score** | |
| Median (range) | 9 (4−13) |
| **World Health Organization (WHO) classification** | |
| Acute myeloid leukemia (AML) with recurrent genetic abnormalities | 10 (10.4%) |
| AML with t (8;21) (q22; q22.1); RUNX1-RUNX1T1 | 4 (4.2%) |
| AML with inv (16) (p13.1q22) or t (16; 16) (p13.1; q22); CBFB-MYH11 | 1 (1.0%) |
| AML with inv (3) (q21.3q26.2) or t (3;3) (q21.3; q26.2); GATA2, MECOM | 1 (1.0%) |
| AML with mutated NPM1 | 4 (4.2%) |
| AML with myelodysplasia-related changes | 24 (25.0%) |
| Therapy-related myeloid neoplasms | 5 (5.2%) |
| AML, not otherwise specified | 57 (59.4%) |
| **Risk groups** | |
| Favorable risk | 9 (9.4%) |
| Intermediate risk | 65 (67.7%) |
| Poor risk | 22 (22.9%) |
| **Bone marrow (BM) blasts** | |
| Median (range) | 56% (20−97) |
| **Peripheral blood (PB) blasts** | |
| Present | 60 (62.5%) |
| Median (range) | 7% (0−92%) |
| **White blood cells, $10^9$/L** | |
| Median (range) | 3.87 (0.51−176.44) |
| **Hemoglobin, g/dL** | |
| Median (range) | 8.3 (3.5−11.9) |
| **Platelet, $10^3$/mm$^3$** | |
| Median (range) | 58 (1−945) |
| **Albumin, g/dL** | |
| Median (range) | 3.7 (2.3−4.8) |
| **Creatinine, mg/dL** | |
| Median (range) | 1.0 (0.4−4.9) |
| **CRP, mg/dL** | |
| Median (range) | 1.75 (0.03−62.5) |
| **Ferritin, ng/mL** | |
| Median (range) | 585 (80−>10000) |

PFS ($P < .001$, hazard ratio [HR] 2.9, 95% CI 1.66−5.07 and $P = .001$, HR 2.54, 95% CI 1.45 −4.44, respectively) and OS ($P = .003$, HR 2.57, 95% CI 1.38−4.8 and $P = .015$, HR 2.2, 95% CI 1.17−4.14, respectively) (Table 4).

**Table 2. Treatment outcomes and adverse events.**

| | |
|---|---|
| **Treatment cycles** | |
| Total | 550 |
| Median (Range) | 4 (1−29) |
| **Response to decitabine** | |
| Complete remission (CR) | 11/85 (12.9%) |
| CR with incomplete blood count recovery (CRi) | 4/85 (4.7%) |
| Partial remission (PR) | 12/85 (14.1%) |
| Hematologic improvement (HI) without an objective response | 18/85 (21.2%) |
| Treatment failure | 58/85 (68.2%) |
| Clinical benefit rate (CR + CRi + PR + HI only) | 45/85 (52.9%) |
| Not evaluable | 11 (11.5%) |
| **Hematologic improvement (HI)** | |
| HI, neutrophil | 29/78 (37.2%) |
| HI, erythrocyte | 32/78 (41.0%) |
| HI, platelet | 28/77 (36.4%) |
| HT, any | 45/90 (50.0%) |
| **Death during induction therapy (during the first cycle of decitabine)** | 12 (12.5%) |
| **Causes of induction mortality** | |
| Infection | 9 (9.4%) |
| Rapidly progressive disease | 2 (2.1%) |
| Other than acute myeloid leukemia (AML) | 1 (1.0%) |
| **Adverse events (AEs) $\geq$ grade 3** | 42 (43.8%) |
| Infection $\geq$ grade 3 | 36 (37.5%) |
| Bacteria $\geq$ grade 3 | 31 (32.3%) |
| Fungus $\geq$ grade 3 | 6 (6.3%) |
| Virus $\geq$ grade 3 | 1 (1.0%) |
| Pneumocystis $\geq$ grade 3 | 1 (1.0%) |
| **Discontinuation of decitabine without progressive disease or treatment-related mortality** | 25 (26.0%) |
| **Causes of discontinuation** | |
| Deteriorated performance | 21 (21.8%) |
| Withdrawal of consent | 2 (2.1%) |
| Unknown | 2 (2.1%) |

## Discussion

In this retrospective analysis, we evaluated 96 older patients with AML who were treated with a decitabine regimen of 5 consecutive days every 4 weeks. The clinical benefit rate (CR + CRi + PR + HI only) was 52.9%, and the median PFS and OS were 7.0 and 10.6 months, respectively. In the previous retrospective studies, reported OS of older AML patients treated with hypomethylating agents were between 8 and 16 months [10–12]. The median OS of our study was comparable to or slightly better than that reported by a pivotal phase III study, DACO-016 (median OS of 7.7 months; 95% CI, 6.2–9.2) [9]. More recently, an Italian multicenter retrospective study including 104 older AML patients treated with decitabine was reported [18]. Seventy-five patients who were received decitabine as first line treatment showed the ORR (CR plus PR) of 42% and median OS of 12.7 months. These results seemed better compared to the result of our study, but they included more patients with good performance status (88% of ECOG performance status 0 or 1). In our study, on the contrary, 52% of patients were ECOG performance status $\geq$ 2, and this proportion of poor performance status might better reflect

**Table 3. Comparison of responders and non-responders to decitabine.**

| | Responders (CR, CRi or PR, n = 27) | Non-responders (n = 69) | Total (n = 96) | *P*-value |
|---|---|---|---|---|
| **Age, years** | | | | .059 |
| Median (Range) | 71.7 (67–87) | 75.0 (65–91) | 73.9 (65–91) | |
| **Sex** | | | | **.022**[*] |
| Male | 21 (77.8%) | 36 (52.2%) | 57 (59.4%) | |
| **Eastern Cooperative Oncology Group (ECOG) performance status** | | | | .349 |
| 0–1 | 15 (55.6%) | 31 (44.9%) | 46 (47.9%) | |
| 2–4 | 12 (44.4%) | 38 (55.1%) | 50 (52.1%) | |
| **Body mass index (BMI)** | | | | .907 |
| Median (Range) | 22.7 (18.5–29.7) | 23.3 (16.3–32.2) | 23.2 (16.3–32.2) | |
| **Mini nutritional assessment short from (MNA-SF) score** | | | | .353 |
| Median (Range) | 10 (4–13) | 9 (4–13) | 9 (4–13) | |
| **World Health Organization (WHO) classification** | | | | .671 |
| AML with recurrent genetic abnormalities | 4 (14.8%) | 6 (8.7%) | 10 (10.4%) | |
| AML with myelodysplasia-related changes | 8 (29.6%) | 16 (23.2%) | 24 (25.0%) | |
| Therapy-related myeloid neoplasms | 1 (3.7%) | 4 (5.8%) | 5 (5.2%) | |
| AML, NOS | 14 (51.9%) | 43 (62.3%) | 57 (59.4%) | |
| **Risk groups** | | | | .227 |
| Favorable | 3 (11.1%) | 6 (8.7%) | 9 (9.4%) | |
| Intermediate | 21 (77.8%) | 44 (63.8%) | 65 (67.7%) | |
| Poor | 3 (11.1%) | 19 (27.5%) | 22 (22.9%) | |
| **Bone marrow (BM) blasts** | | | | **.015**[*] |
| Median (Range) | 43% (20–90) | 62% (20–97) | 56% (20–97) | |
| **Peripheral blood (PB) blasts** | | | | |
| Present | 12 (44.4%) | 48 (69.6%) | 60 (62.5%) | **.022**[*] |
| Median (Range) | 0% (0–84%) | 10% (0–92%) | 7% (0–92%) | .204 |
| **White blood cells, $10^9$/L** | | | | .216 |
| Median (Range) | 2.89 (0.83–87.99) | 4.12 (0.51–176.44) | 3.87 (0.51–176.44) | |
| **Hemoglobin, g/dL** | | | | .757 |
| Median (Range) | 8.5 (4.7–11.4) | 8.2 (3.5–11.9) | 8.3 (3.5–11.9) | |
| **Platelet, $10^3$/mm$^3$** | | | | .248 |
| Median (Range) | 56 (10–180) | 61 (1–945) | 58 (1–945) | |
| **Albumin, g/dL** | | | | **.010**[*] |
| Median (Range) | 3.9 (3.0–4.8) | 3.7 (2.3–4.7) | 3.7 (2.3–4.8) | |
| **Creatinine, mg/dL** | | | | .701 |
| Median (Range) | 1.15 (0.5–1.81) | 0.92 (0.4–4.9) | 1.0 (0.4–4.9) | |
| **CRP, mg/dL** | | | | **.004**[*] |
| Median (Range) | 0.68 (0.03–21.43) | 2.62 (0.05–62.5) | 1.75 (0.03–62.5) | |
| **Ferritin, ng/mL** | | | | .547 |
| Median (Range) | 585 (80–7803) | 585 (90–>10000) | 585 (80–>10000) | |

[*] Statistically significant *P* values are shown in bold.

the reality of the actual practice of elderly AML. Twelve patients (12.5%) died during the first cycle of decitabine treatment, and most induction mortalities were caused by infection (n = 9). Since 25 patients (26.0%) discontinued decitabine before disease progression or death, the median number of treatment cycles was only 4. High rates of early mortality and discontinuation of treatment without disease progression might be a reflection of a real-world practice,

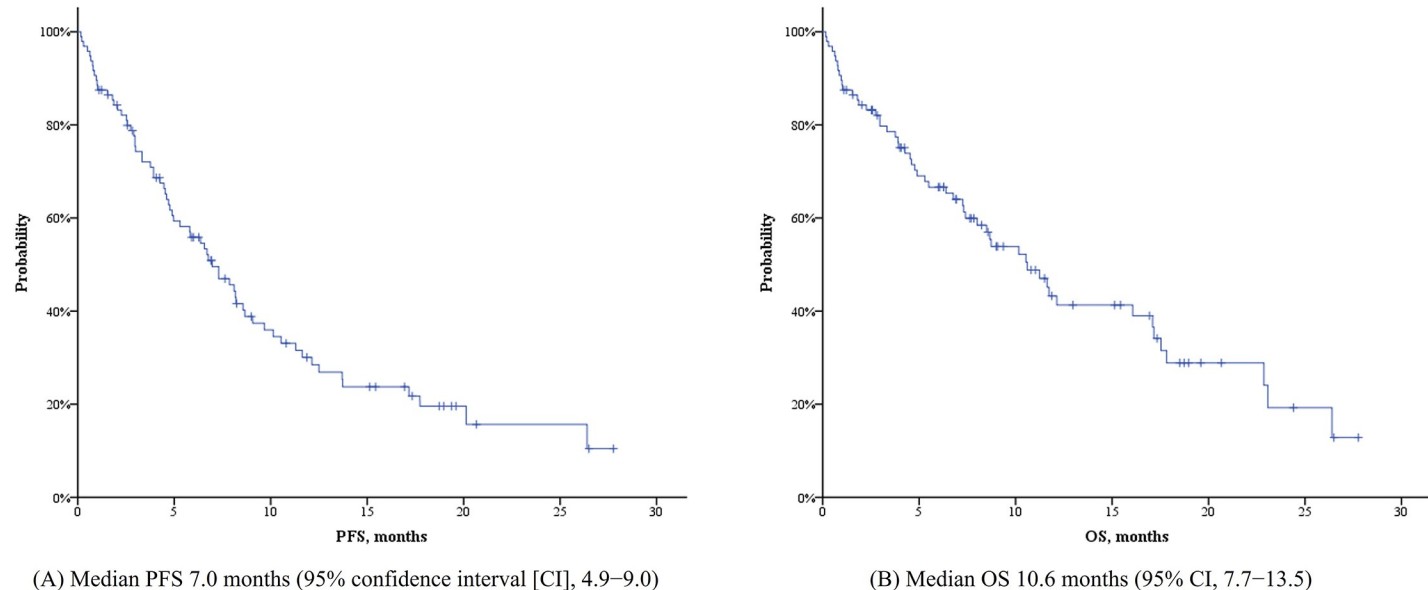

(A) Median PFS 7.0 months (95% confidence interval [CI], 4.9−9.0)

(B) Median OS 10.6 months (95% CI, 7.7−13.5)

**Fig 1.** Progression-free survival (PFS, A) and overall survival (OS, B). (A) Median PFS 7.0 months (95% confidence interval [CI], 4.9−9.0) (B) Median OS 10.6 months (95% CI, 7.7−13.5).

emphasizing the importance of appropriate selection of candidates for decitabine treatment. An increase in the experience of clinicians regarding use of hypomethylating agents and sophisticated management of adverse events, including infections, would improve the clinical outcomes of those patients.

Both patient-related and disease-related factors are considered when selecting the treatment intensity of older AML patients [19]. Performance status, functional status, and comorbid conditions are important for treatment with hypomethylating agents as well as induction chemotherapy. The importance of geriatric assessment including nutritional status has been emphasized for a long time, and the use of an MNA-SF has been suggested as a useful tool for

**Table 4.  Multivariate analysis for Progression-Free Survival (PFS) and Overall Survival (OS).**

| | PFS | | | | OS | | | |
|---|---|---|---|---|---|---|---|---|
| | Univariate *P* | Multivariate *P* | Hazard ratio (HR) | 95% confidence interval (CI) | Univariate *P* | Multivariate *P* | HR | 95% CI |
| **Age (≤74 vs. >75)** | 0.028 | | | | 0.001 | **0.026**[*] | 1.9 | 1.08 −3.34 |
| **ECOG PS**[†] **(0–1 vs. 2–4)** | < .001 | | | | < .001 | | | |
| **MNA-SF**[‡] **score (normal to at risk vs. poor)** | < .001 | **< .001** | 2.9 | 1.66−5.07 | < .001 | **0.003**[*] | 2.57 | 1.38 −4.8 |
| **PB**[§] **blasts (absence vs. presence)** | < .001 | **0.001** | 2.54 | 1.45−4.44 | 0.001 | **0.015**[*] | 2.2 | 1.17 −4.14 |
| **Cytogenetic risk (favorable or intermediate vs. poor)** | 0.007 | **0.032** | 1.86 | 1.06−3.28 | 0.096 | | | |

[†]Eastern Cooperative Oncology Group performance status

[‡]Mini nutritional assessment short from

[§]Peripheral blood.

[*]Statistically significant *P* values are shown in bold.

assessing older patients' nutritional status [20–22]. Although various validated tools for geriatric assessment of AML patients were introduced [23], they have not been uniformly applied in clinical practice due to the diversity and complexity of the tools. Meanwhile, the MNA-SF consists of only 6 components pertaining to information about nutritional status, active daily living, psychological stress, and active disease. Thus, it provides a simple and quick method for identifying who is at risk of malnutrition, or who is already malnourished, in combination with the general status of older patients [13].

In the multivariate and subgroup analyses of a prior phase III study, Cox proportional hazards model was used for evaluating effects of various factors including age, sex, cytogenetic risk (intermediate vs. poor), type of AML (de novo vs. secondary), ECOG performance status (0 or 1 vs. 2), BM blasts (>50% vs. ≤50%), baseline platelets, and white blood cells on OS and response rates [9, 24]. Since this study was a multinational trial, geographic region was also used as a parameter in the analysis. Meanwhile, in our study, we assessed baseline nutritional status using MNA-SF. Both ECOG performance status and the MNA-SF score were associated with prolonged measures of survival, as determined by the subgroup analysis. BM blasts were not statistically significant for survival in univariate subgroup analysis with cut-off values of 30% or 50%. Instead of BM blasts, the absence of PB blasts had a positive impact on PFS and OS. Thus, 5 covariates (age, ECOG performance status, MNA-SF score, the absence of PB blasts, and cytogenetic risk) were included in multivariate Cox regression for PFS and OS. Finally, it was revealed that the MNA-SF score was the most significant factor for predicting both PFS and OS. The MNA-SF is a convenient and effective tool for predicting measures of survival of older AML patients who were treated with hypomethylating agents, and large-scale prospective studies are needed to confirm the role of MNA-SF for geriatric assessment prior to the initiation of treatment of AML.

Focusing on 9 patients with favorable risk (4 patients with t(8;21), 1 patient with inv(16), and 4 patients with mutated NPM1), 3 patients (33.3%) achieved CR or PR and 2 (22.2%) showed HI without an objective response. The median PFS of the favorable risk group was 8.2 months (95% CI, 3.2–13.2), and the median OS was not reached. Currently, intensive chemotherapy is generally recommended for older patients with favorable cytogenetics [15, 25]. There were insufficient data on the use of hypomethylating agents in AML with favorable risk. However, anecdotal evidence of long-term responders to decitabine without induction chemotherapy or hematopoietic cell transplantation was reported, and one such patient had core-binding factor (CBF) with t(8;21) [26]. Similarly, one patient enrolled in our study who had CBF with t(8;21) has long-term CR and will receive more than 40 cycles of decitabine in 2019. Thus, further studies comparing hypomethylating agents to intensive chemotherapy in older patients with favorable cytogenetics are warranted.

In this study, it was observed that the absence of PB blasts was associated with a better response to decitabine and longer PFS and OS. The favorable effect of the absence of PB blasts continued to multivariate analysis for both PFS and OS. Similarly, other retrospective studies with hypomethylating agents also reported the association of higher PB blasts and poor survival outcomes [10, 27]. DiNardo et al. demonstrated that younger AML patients (≤ 60 years) receiving intensive chemotherapy showed similar outcomes regardless of their BM blast percentage, whereas older patients (≥ 70 years) with 20–29% blasts had outcomes similar to that of patients with < 20% blasts and better outcomes than those with ≥ 30% blasts in their BM [28]. It is thought that more advanced disease with a high blast count in either the PB or BM had a negative impact on treatment outcomes, especially in older patients with AML.

This study has several limitations. All data were collected in a retrospective manner, and the MNA-SF score was also calculated retrospectively by matching patients' data from EMRs to the parameters of the MNA-SF. Although all participating institutions had collected all

MNA-SF-related indicators at the time of diagnosis of AML in a routine procedure, the greatest limitation of this study is the reliability of MNA-SF because detailed information may have been collected differently for the purpose of the study. Furthermore, MNA-SF was assessed only once at the time of diagnosis, and changes in the course of decitabine treatment could not be evaluated. Thus, the value of a dynamic assessment for nutritional status using MNA-SF in older patients of AML needs to be reconfirmed in future prospective studies. The percentage of blasts in the BM and PB were collected from the laboratory reports at each site, without the undertaking of a central review of specimens. However, this study analyzed 96 AML patients. This is one of the largest retrospective studies that enrolled an older Asian population of AML patients who were treated with decitabine.

In conclusion, the current study suggested that decitabine demonstrated acceptable treatment outcomes in older patients with AML. In this population, the MNA-SF score was the most valuable predictor for the response and outcomes, and the absence of PB blasts was also associated with improved measures of survival. Further studies are warranted to develop a prognostic model for decitabine treatment, with a greater focus on geriatric and nutritional perspectives.

## Supporting information

**S1 Fig. . Subgroup analyses for Progression-Free Survival (PFS) by age ($<$ 75 vs. $\geq$ 75 years), Eastern Cooperative Oncology Group (ECOG) performance status (ECOG 0–1 vs. 2–4), cytogenetic risk (favorable or intermediate risk vs. poor risk), peripheral blood blasts (absence vs. presence), and mini nutritional assessment short from (MNA-SF) score ($\geq$ 8 vs. $<$ 8).**
(TIF)

**S2 Fig. Subgroup analyses for Overall Survival (OS) by age ($<$ 75 vs. $\geq$ 75 years), Eastern Cooperative Oncology Group (ECOG) performance status (ECOG 0–1 vs. 2–4), cytogenetic risk (favorable or intermediate risk vs. poor risk), peripheral blood blasts (absence vs. presence), and Mini Nutritional Assessment Short From (MNA-SF) score ($\geq$ 8 vs. $<$ 8).**
(TIF)

## Acknowledgments

This research was conducted under the auspices of the Korean Society of Hematology (KSH), Gyeonggi/Incheon Branch. We thank all centers that participated in this study.

## Author Contributions

**Conceptualization:** Junshik Hong, Seong Kyu Park.

**Data curation:** Kwai Han Yoo, Jinhyun Cho, Boram Han, Se Hyung Kim, Dong-Yeop Shin, Junshik Hong, Hawk Kim, Hyo Jung Kim, Dae young Zang, Sung-Soo Yoon, Jong-Youl Jin, Jae Hoon Lee, Dae-Sik Hong, Seong Kyu Park.

**Formal analysis:** Kwai Han Yoo, Junshik Hong.

**Writing – original draft:** Kwai Han Yoo.

**Writing – review & editing:** Kwai Han Yoo, Junshik Hong, Seong Kyu Park.

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
