## [Decision Letter · Decision Letter 0]

12 Feb 2020

PONE-D-20-00626

Outcomes of decitabine treatment for newly diagnosed acute myeloid leukemia in older adults and the role of a mini nutritional assessment short form

PLOS ONE

Dear Dr. Hong,

Thank you for submitting your manuscript to PLOS ONE. After careful consideration, we feel that it has merit but does not fully meet PLOS ONE’s publication criteria as it currently stands. Therefore, we invite you to submit a revised version of the manuscript that addresses the points raised during the review process by both Reviewers, experts in the field.

We would appreciate receiving your revised manuscript by Mar 28 2020 11:59PM. To enhance the reproducibility of your results, we recommend that if applicable you deposit your laboratory protocols in protocols.io, where a protocol can be assigned its own identifier (DOI) such that it can be cited independently in the future. For instructions see: http://journals.plos.org/plosone/s/submission-guidelines#loc-laboratory-protocols

We look forward to receiving your revised manuscript.

Kind regards,

Francesco Bertolini, MD, PhD

Academic Editor

PLOS ONE

Journal Requirements:

2. In the ethics statement in the manuscript and in the online submission form, please provide additional information about the patient records used in your retrospective study, including: a) whether all data were fully anonymized before you accessed them; b) the date range (month and year) during which patients' medical records were accessed and c) the source of the medical records analyzed in this work (e.g. the names of the six institutions used in this study).

4. Please include your tables as part of your main manuscript and remove the individual files. Please note that supplementary tables should remain as separate "supporting information" files.

Reviewers' comments:

Reviewer's Responses to Questions

**Comments to the Author**

1. Is the manuscript technically sound, and do the data support the conclusions?

Reviewer #1: Yes

Reviewer #2: Yes

2. Has the statistical analysis been performed appropriately and rigorously? 

Reviewer #1: Yes

Reviewer #2: Yes

3. Have the authors made all data underlying the findings in their manuscript fully available?

Reviewer #1: Yes

Reviewer #2: No

4. Is the manuscript presented in an intelligible fashion and written in standard English?

Reviewer #1: Yes

Reviewer #2: Yes

5. Review Comments to the Author

Reviewer #1: Thank you for the opportunity to review this article. This article is well written and deals with an interesting topic on real-life assessment and treatment with decitabine. Subsequently, the authors tried to evaluate prognostic factors.

The major problem is the construction of the article, which oscillates between real-life study and the identification of prognostic factors. This can be seen from the title to the discussion.

The prognostic factor MNA-SF that is put forward is too unreliable in its mode of evaluation. I will focus more on real-life data and secondly on prognostic factors in the minor axis.

TITLE : The major point of the publication is the evaluation of the survival of patients on decitabine in real world and it should be highlighted in the title and not the identification of prognostic factors

The title should not include prognostic factors.

Introduction

Indications of decitabine for older adults with AML are still controversial. You must insist on this point

the United States Food and Drug Administration did not approve decitabine for the treatment of newly diagnosed AML based on the results of a clinical trial reported by Kantarjian, et al

the European Medicines Agency Committee for Medicinal Products for Human Use approved decitabine as a first-line treatment for older adults with AML as the Korean Food and Drug Administration (KFDA).

Your objectif was to analyse outcomes in real world of decitabine treatment in older patients with newly diagnosed. It shoul be clearly define.

Do not put a reference (13) in your objective. The notion of nutritional status should not to be introduced int this part because it is not your main objective.

Patients and Methods

How were you able to do the MNA-SF retrospectively ? how do you find the data of "Acute illness or psychological stress in the last 3 months" or "Has the patient eaten less in the last 3 months due to lack of appetite, digestive problems, chewing or swallowing difficulties? "

Give more tell about the factors that we included in multivariate analysis. Please detail the multivariate statistical analyses and explain cut-off of MNA-SF.

Most of time : Baseline bone marrow blast, Baseline platelets and / WBC were included in multivariate analysis

Resutls

I don’t understand this sentence : “Forty-five patients (45/90, 50.0%) showed HI in absolute neutrophil count (29/78, 37.2%)” because you describe 18 patients with HI. I think you should rephrase or add punctuation.

Explain in materiel and methods your definition of “infectious complications” . It included bacterial infections and fungal infection complications?

Discussion

The most limiting point is the identification of the MNA-SF, which is unreliable. Its identification as a prognostic factor must be taken with caution.

You do not enough compare your results (cut-offs, results) with other existing publications (10-12) and I added news references :

“Decitabine as a First-Line Treatment for Older Adults Newly Diagnosed with Acute Myeloid Leukemia” Hyunsung Park et al.

“Decitabine improves outcomes in older patients with acute myeloid leukemia and higher blast counts“.

Kadia et al.

“Multivariate and subgroup analyses of a randomized, multinational, phase 3 trial of decitabine vs treatment choice of supportive care or cytarabine in older patients with newly diagnosed acute myeloid leukemia and poor- or intermediate-risk cytogenetics.”

Mayer J et al.

Figures

Poor quality figures due to a pixel problem, impossible to read

Reviewer #2: 1. The authors stated in the “treatment and evalution paragraph” that “considering the palliative nature of the HMA treatment, disease evalutation was often not performed unless clear signes of disease progression”.

I think that treatment with HMA is not a palliative treatment but an active therapy for elderly AML that can be berfomed when the patient is considered unfit for aggressive therapy; Unfit patient must be defined based on a multidimentional geriatric assesment that ditiguish them clearly from a frail patient for which best supportive care is suggested.

In absence of a systematic disease evaluation for all patient population with bone marrow aspirate/biopsy how could you determine properly disease response ( es CR/CRi)?

In Table 1 which describes the patients characteristics their multidimentional geriatric assesment is missing. Thus how did you decided that patients were unfit for aggressive chemotherapy or frail? I do not think that ECOG performance status assesment in sufficient for a correct fitness evaluation

2. The highlight and the novelty of the study is that MNA-SF assessment ≥8 predicts improved survival in decitabine treated AML patients.

I think that it should be specified when MNA-SF score was performed. If was performed at diagnosis and /or every how many months durnig decitabine treatment.

Infact a dynamic assesment is more significative because the score value can chage a lot along tratment and on the base of disease response to therapy

6. PLOS authors have the option to publish the peer review history of their article (what does this mean?). If published, this will include your full peer review and any attached files.

Reviewer #1: No

Reviewer #2: No

---

## [Author Response · Author response to Decision Letter 0]

6 Apr 2020

Reviewer #1:

Thank you for the opportunity to review this article. This article is well written and deals with an interesting topic on real-life assessment and treatment with decitabine. Subsequently, the authors tried to evaluate prognostic factors.

The major problem is the construction of the article, which oscillates between real-life study and the identification of prognostic factors. This can be seen from the title to the discussion.

The prognostic factor MNA-SF that is put forward is too unreliable in its mode of evaluation. I will focus more on real-life data and secondly on prognostic factors in the minor axis.

TITLE : The major point of the publication is the evaluation of the survival of patients on decitabine in real world and it should be highlighted in the title and not the identification of prognostic factors.

The title should not include prognostic factors.

 We changed the title to “Real-world outcomes of decitabine treatment for newly diagnosed acute myeloid leukemia in older adults”.

Introduction

Indications of decitabine for older adults with AML are still controversial. You must insist on this point the United States Food and Drug Administration did not approve decitabine for the treatment of newly diagnosed AML based on the results of a clinical trial reported by Kantarjian, et al.

The European Medicines Agency Committee for Medicinal Products for Human Use approved decitabine as a first-line treatment for older adults with AML as the Korean Food and Drug Administration (KFDA).

Your objective was to analyze outcomes in real world of decitabine treatment in older patients with newly diagnosed. It should be clearly defined.

Do not put a reference (13) in your objective. The notion of nutritional status should not to be introduced int this part because it is not your main objective.

 We added controversies about the use of decitabine for the first-line treatment of AML. A sentence and a reference about MNA-SF in the part of introduction were removed following the reviewers’ recommendations.

Patients and Methods

How were you able to do the MNA-SF retrospectively? how do you find the data of "Acute illness or psychological stress in the last 3 months" or "Has the patient eaten less in the last 3 months due to lack of appetite, digestive problems, chewing or swallowing difficulties?"

 All institutions participating in this study had acquired all MNA-SF-related indicators at the time of diagnosis of AML, through hospitalization records, baseline nursing records, and nutritional records by on-site nutritionists. We could access the data about "Acute illness and psychological stress in the last 3 months" in hospitalization records and baseline nursing records, and the data about "Has the patient eaten less in the last 3 months due to lack of appetite, digestive problems, chewing or swallowing difficulties?" in baseline nursing records, and nutritional records.

Give more tell about the factors that we included in multivariate analysis. Please detail the multivariate statistical analyses and explain cut-off of MNA-SF.

Most of time : Baseline bone marrow blast, Baseline platelets and / WBC were included in multivariate analysis

 In multivariate analysis by Cox regression model, we used 5 covariates (age, ECOG PS, MNA-SF score, the absence of PB blasts, and cytogenetic risk). Four of them (Age, ECOG PS, MNA-SF score, and the absence of PB blasts) showed significance in univariate analyses of both PFS and OS. Cytogenetic risk (favorable or intermediate risks vs. poor risk) which was statistically associated with improved PFS (P = .007), but not OS (P = .096), was also added in the model. Baseline bone marrow blasts percentage and platelet count were not associated with survivals in our study population. Therefore, we did not use those factors in multivariate analysis.

Results

I don’t understand this sentence : “Forty-five patients (45/90, 50.0%) showed HI in absolute neutrophil count (29/78, 37.2%)” because you describe 18 patients with HI. I think you should rephrase or add punctuation.

 Those 18 patients did not achieve an objective response (CR or PR), but they showed hematologic improvements (His). We revised the sentences more clearly as following: “18 patients (21.2%) who did not achieve an objective response demonstrated hematologic improvement (HI) in PB. Thus, the clinical benefit rate (CR + CRi + PR + HI only) was 52.9% (45/85). Regardless of achieving an objective response, 45 patients (45/90, 50.0%) showed HI in absolute neutrophil count (29/78, 37.2%), hemoglobin (32/78, 41.0%) and/or platelet count (28/77, 36.4%).”

Explain in materiel and methods your definition of “infectious complications”. It included bacterial

infections and fungal infection complications?

 We added a sentence, “Infectious complications included any bacterial, viral, fungal, and miscellaneous infection such as Pneumocystis jiroveci during decitabine treatment.” in the Patients and Methods section.

Discussion

The most limiting point is the identification of the MNA-SF, which is unreliable. Its identification as a prognostic factor must be taken with caution.

 We accepted your suggestions and emphasized the limitations of the use of MNA-SF by retrospective measures. We also mentioned the need for prospective studies for confirming the value of MNA-SF in elderly AML.

You do not enough compare your results (cut-offs, results) with other existing publications (10-12) and I added new references:

“Decitabine as a First-Line Treatment for Older Adults Newly Diagnosed with Acute Myeloid Leukemia” Hyunsung Park et al.

“Decitabine improves outcomes in older patients with acute myeloid leukemia and higher blast counts“ Kadia et al.

“Multivariate and subgroup analyses of a randomized, multinational, phase 3 trial of decitabine vs

treatment choice of supportive care or cytarabine in older patients with newly diagnosed acute myeloid leukemia and poor- or intermediate-risk cytogenetics.” Mayer J et al.

 Many thanks for the list of existing publications. We added a comparison of survival and cut-offs and reason for including PB blasts in multivariate analysis rather than BM blasts in the part of discussion.

Figures

Poor quality figures due to a pixel problem, impossible to read

 The resolution of original figures satisfies the guidelines. The Supplement figures shown in the PDF are too small, so it is difficult to see. 

You can download the original files from the link at the top right of figure pages in the PDF.

 

Reviewer #2:

1. The authors stated in the “treatment and evaluation paragraph” that “considering the palliative nature of the HMA treatment, disease evaluation was often not performed unless clear signs of disease progression”.

I think that treatment with HMA is not a palliative treatment but an active therapy for elderly AML that can be performed when the patient is considered unfit for aggressive therapy; Unfit patient must be defined based on a multidimensional geriatric assessment that distinguish them clearly from a frail patient for which best supportive care is suggested.

 We intended “palliative nature” as a meaning of different mechanism and responses from traditional intensive chemotherapy. As the reviewer pointed out, the meaning of “palliative nature” is unclear, so we corrected this phrase as follows: “considering the mechanism of action of decitabine and delayed responses different from intensive chemotherapy,”

In absence of a systematic disease evaluation for all patient population with bone marrow aspirate/biopsy how could you determine properly disease response ( es CR/CRi)?

 Patients with new presence of leukemic blasts in peripheral blood or lack/loss of hematologic improvement during treatment were thought of disease progression or no response. In that cases, bone marrow (BM) biopsy were not needed to confirm CR or PR. This study was a retrospective study, and it reflected a real-world practice. However, every patients with CR or PR in our study were confirmed by BM biopsy.

In Table 1 which describes the patients characteristics their multidimensional geriatric assessment is missing. Thus how did you decided that patients were unfit for aggressive chemotherapy or frail? I do not think that ECOG performance status assessment in sufficient for a correct fitness evaluation

 This study is a retrospective study from multicenter. Thus, a uniform multidimensional geriatric assessment was not performed, and physicians of each center individually decided to start decitabine treatment. This might reflect the actual situation of our clinical practice.

2. The highlight and the novelty of the study is that MNA-SF assessment ≥8 predicts improved survival in decitabine treated AML patients.

I think that it should be specified when MNA-SF score was performed. If was performed at diagnosis and/or every how many months during decitabine treatment.

In fact a dynamic assessment is more significative because the score value can change a lot along treatment and on the base of disease response to therapy

 All the MNA-SF-related indicators were collected only once at the time of diagnosis of AML. We absolutely agreed with the reviewer’s point-out, and we discussed more about necessity of a dynamic assessment of MNA-SF, by prospective approach.

---

## [Decision Letter · Decision Letter 1]

22 Apr 2020

PONE-D-20-00626R1

Real world outcomes of decitabine treatment for newly diagnosed acute myeloid leukemia in older adults

PLOS ONE

Dear Dr. Hong,

Thank you for submitting your manuscript to PLOS ONE. After careful consideration, we feel that it has merit but does not fully meet PLOS ONE’s publication criteria as it currently stands. Therefore, we invite you to submit a revised version of the manuscript that addresses all the points raised during the review process by Reviewer #1, an expert in the field.

We would appreciate receiving your revised manuscript by Jun 06 2020 11:59PM. To enhance the reproducibility of your results, we recommend that if applicable you deposit your laboratory protocols in protocols.io, where a protocol can be assigned its own identifier (DOI) such that it can be cited independently in the future. For instructions see: http://journals.plos.org/plosone/s/submission-guidelines#loc-laboratory-protocols

We look forward to receiving your revised manuscript.

Kind regards,

Francesco Bertolini, MD, PhD

Academic Editor

PLOS ONE

Reviewers' comments:

Reviewer's Responses to Questions

**Comments to the Author**

1. If the authors have adequately addressed your comments raised in a previous round of review and you feel that this manuscript is now acceptable for publication, you may indicate that here to bypass the “Comments to the Author” section, enter your conflict of interest statement in the “Confidential to Editor” section, and submit your "Accept" recommendation.

Reviewer #1: All comments have been addressed

Reviewer #2: All comments have been addressed

2. Is the manuscript technically sound, and do the data support the conclusions?

Reviewer #1: Partly

Reviewer #2: Yes

3. Has the statistical analysis been performed appropriately and rigorously? 

Reviewer #1: No

Reviewer #2: Yes

4. Have the authors made all data underlying the findings in their manuscript fully available?

Reviewer #1: Yes

Reviewer #2: Yes

5. Is the manuscript presented in an intelligible fashion and written in standard English?

Reviewer #1: Yes

Reviewer #2: Yes

6. Review Comments to the Author

Reviewer #1: Article that has improved since the last time, but there are still points that need to be clarified.

Materiels and Methods :

Change title

Definition of real word : Real world data (RWD) in medicine is data derived from a number of sources that are associated with outcomes in a heterogeneous patient population in real-world settings, such as patient surveys, clinical trials, and observational cohort studies.[1] Real-world data refer to observational data as opposed to data gathered in an experimental setting such as a randomized controlled trial (RCT).

Remove real word in the text

I am very dubious about the retrospective evaluation of the MNA-SF, which for me will inevitably be subjective due to its retrospective character and therefore unreliable. You say that the people have partly made a prospective and retrospective evaluation, so why didn't you do the questionnaire from the beginning ?

You said that the classification was done according to the 2016 classification, but what about before 2016? Did you reclassify the AML?

The detail of the univariate/multivariate analysis are missing in methods: which criteria, which p...

PFS and OS should be describe in statistical analysis

Results

The description of the results on prognostic factors, the most important one, is confusing:

for example: "age, PS, absence of PB... were associated to prolonged OS"

Does it involve univariate, multivariate analysis?

Figures: p and months are missing

Tables : HR and 95%CI are missing in univariate analysis

Discussion

Difficult in the discussion to see the original points of the article with numbers digits of the publication appearing.

In a recent publication (Fili et al. 2019, leukemia research) found : The ORR in real-world setting was 33% and median OS in responders was 22.6 mths. Which is more than you, how can you explain the difference ?

Reviewer #2: The authors have adequately addressed my comments raised in a previous round of review and you feel that this manuscript is now acceptable for publication,

7. PLOS authors have the option to publish the peer review history of their article (what does this mean?). If published, this will include your full peer review and any attached files.

Reviewer #1: No

Reviewer #2: No

---

## [Author Response · Author response to Decision Letter 1]

5 Jun 2020

Reviewer #1: Article that has improved since the last time, but there are still points that need to be clarified.

Materiels and Methods :

Change title

Definition of real word : Real world data (RWD) in medicine is data derived from a number of sources that are associated with outcomes in a heterogeneous patient population in real-world settings, such as patient surveys, clinical trials, and observational cohort studies.[1] Real-world data refer to observational data as opposed to data gathered in an experimental setting such as a randomized controlled trial (RCT).

Remove real word in the text

 We removed “real world” in the title and introduction section.

I am very dubious about the retrospective evaluation of the MNA-SF, which for me will inevitably be subjective due to its retrospective character and therefore unreliable. You say that the people have partly made a prospective and retrospective evaluation, so why didn't you do the questionnaire from the beginning?

 MNA-SF-related factors were obtained by physicians, nurses, and nutritionists in participating institutions. They were recorded on hospitalization records, baseline nursing records, and nutritional records for purposes unrelated to this study. As reviewers pointed out, MNA-SF scores were retrospectively calculated by reviewing medical records, therefore the reliability of MNA-SF score is inevitably lower than that of prospective studies. Since the predictive power of this score was expected to be weak, it was only described as an additional content according to recommendations. We also described this limitation in discussion section.

You said that the classification was done according to the 2016 classification, but what about before 2016? Did you reclassify the AML?

 AML was initially categorized by WHO classification of myeloid neoplasms and acute leukemia at the time of diagnosis in each patient (i.e., either version 2008 or 2016). We reclassified the category according to the 2016 revision of WHO classification when we obtained and analyzed these data.

The detail of the univariate/multivariate analysis are missing in methods: which criteria, which p...

 We added detailed description about the methods of analysis in statistical analysis section.

PFS and OS should be described in statistical analysis.

 Definition of PFS and OS was moved to statistical analysis section from treatment and evaluation section.

Results

The description of the results on prognostic factors, the most important one, is confusing:

for example: "age, PS, absence of PB... were associated to prolonged OS"

Does it involve univariate, multivariate analysis?

 As a reviewer pointed out, methods of statistics were unclear. We added a clear explanation in methods section and tailed additional markers (univariate/multivariate) in the sentences of result section. 

Figures: p and months are missing

 Figure 1A and 1B: Survival duration (months) and 95% CI were already presented in Figure legends.

 Figure S1 and S2: P-value was given in lower left corner of each graph. We added survival duration (months) and 95% CI of each curve.

Tables: HR and 95%CI are missing in univariate analysis

 Univariate analysis of survival was conducted by using log-rank test. Thus, univariate P was presented in Table 4, but hazard ratio cannot be calculated in univariate analysis.

Discussion

Difficult in the discussion to see the original points of the article with numbers digits of the publication appearing.

 We expressed the exact number of survivals from other studies more clearly (reference 9-12, 18) to compare with the results of ours.

In a recent publication (Fili et al. 2019, leukemia research) found : The ORR in real-world setting was 33% and median OS in responders was 22.6 months. Which is more than you, how can you explain the difference?

 In the publication (Fili et al. 2019, leukemia research), the ORR was 42% and median OS was 12.7 months in patients treated with first line decitabine (n = 75). These numbers seemed better than our results, but they involved more proportion of better performance status (ECOG PS 1-2 was 88%) than ours. We added explanation about this in the first paragraph of discussion section.

---

## [Editor Report · Decision Letter 2]

17 Jun 2020

Outcomes of decitabine treatment for newly diagnosed acute myeloid leukemia in older adults

PONE-D-20-00626R2

Dear Dr. Hong,

We’re pleased to inform you that your manuscript has been judged scientifically suitable for publication and will be formally accepted for publication once it meets all outstanding technical requirements.

Kind regards,

Francesco Bertolini, MD, PhD

Academic Editor

PLOS ONE
---

## [Editor Report · Acceptance letter]

23 Jul 2020

PONE-D-20-00626R2 

Outcomes of decitabine treatment for newly diagnosed acute myeloid leukemia in older adults 

Dear Dr. Hong:

I'm pleased to inform you that your manuscript has been deemed suitable for publication in PLOS ONE. Congratulations! Your manuscript is now with our production department. 

Kind regards, 

on behalf of

Dr. Francesco Bertolini 

Academic Editor

PLOS ONE